# Reinforcement Learning with Maskable Stock Representation for Portfolio Management in Customizable Stock Pools

## ABSTRACT

Portfolio management (PM) is a fundamental financial trading task, which explores the optimal periodical reallocation of capitals into different stocks to pursue long-term profits. Reinforcement learning (RL) has recently shown its potential to train profitable agents for PM through interacting with financial markets. However, existing work mostly focuses on fixed stock pools, which is inconsistent with investors' practical demand. Specifically, the target stock pool of different investors varies dramatically due to their discrepancy on market states and individual investors may temporally adjust stocks they desire to trade (e.g., adding one popular stocks), which lead to customizable stock pools (CSPs). Existing RL methods require to retrain RL agents even with a tiny change of the stock pool, which leads to high computational cost and unstable performance. To tackle this challenge, we propose `EarnMore`, a rEinforcement leARNing framework with Maskable stOck REpresentation to handle PM with CSPs through one-shot training in a global stock pool (GSP). Specifically, we first introduce a mechanism to mask out the representation of the stocks outside the target pool. Second, we learn meaningful stock representations through a self-supervised masking and reconstruction process. Third, a re-weighting mechanism is designed to make the portfolio concentrate on favorable stocks and neglect the stocks outside the target pool. Through extensive experiments on 8 subset stock pools of the US stock market, we demonstrate that `EarnMore` significantly outperforms 14 state-of-the-art baselines in terms of 6 popular financial metrics with over 40% improvement on profit.

## KEYWORDS

Portfolio Management, Reinforcement Learning, Representation Learning

## 1 INTRODUCTION

The stock market, which involves over $90 trillion market capitalization, has attracted the attention of innumerable investors around the world. Portfolio management, which dynamically allocates the proportion of capitals among different stocks, plays a key role to make profits for investors. Reinforcement learning (RL) has recently become a promising methodology for financial trading tasks due to its stellar performance on solving complex sequential decision-making problems such as Go [25] and matrix multiplication [9]. In fact, RL has achieved significant success in various quantitative trading tasks such as algorithmic trading [4], portfolio management [29], order execution [8] and market making [26]. To apply RL methods for PM, existing work [29, 30, 33] always train RL agents to make investment decisions based on a fixed stock pool[1], which requires retraining with high computational cost when investors need to change their target stock pools.

---

[1]The fixed stock pool in existing work is the global stock pool (GSP) under our setting.

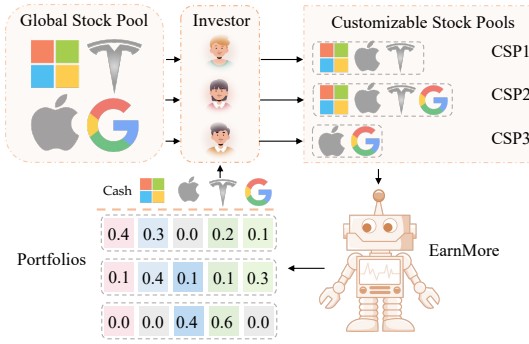

**Figure 1: Overview of portfolio management by `EarnMore` in customizable stock pools (CSPs).**

Furthermore, investors are unable to engage in or influence the agent's decision-making process, as it is uncontrollable for them. Investors and agents may work better with each other in collaboration. On one hand, investors have a limited understanding of stock trends, hidden connections among stocks, and the overall market dynamics. On the other hand, the RL agent may make occasional decision errors and lacks the human capacity to acquire information in diverse ways. All of these factors can result in diminished or suboptimal returns. For instance, if a certain stock is delisted in the trading process, the agent may still consider it as a candidate for investment allocation, potentially leading to a loss of returns. In a different scenario, when a stock exhibits significant future profit potential, investors may want to add it to the target pools to maximize their returns. Therefore, We introduce the task of portfolio management with customizable stock pools (CSPs) to meet the mentioned demands and address the above issues.

As shown in Fig. 1, the demand of PM with customizable stock pools (CSPs) is ubiquitous for different financial practitioners in real-world trading scenarios. For instance, stock brokerages need to offer real-time portfolio suggestions for millions of investors with diversified preferences on stock pools. The investors also desire to adapt their stock pools based on different market conditions from time to time. Therefore, an RL algorithm with the ability to handle PM with CSPs is urgently needed. There are 3 straightforward methods for implementing PM in CSPs: i) training an agent from scratch on each individual CSP. However, randomly picking 5 stocks from 30 for a CSP leads to 140k+ combinations, which is unfeasible in practice; ii) adjusting the output dimensions of the policy network and then fine-tune the agents by mapping the action space of the GSP to CSP. Fine-tuning agent on PM with contiguous action space might be equally time-consuming as starting from scratch, making it impractical in reality; iii) using action-masking method by subtracting a large constant from policy network logits that represent unfavorable stocks to reduce their investment allocation. Human-induced decision changes in this method do not truly express the agent's real decision-making, as it lacks awareness of CSP

stocks, significantly impacting effectiveness. Although the existing work considers the dynamics of the stock pool, such as the study by Betancourt et al. [2] focuses on the changing number of assets, training an actor and critic for each stock is resource-intensive and time-consuming. Additionally, this approach disallows for changes in the stock pool in terms of number of stocks and composition during the investment process, which would suffer the same limitations if implemented by action-masking.

To handle PM with CSPs, we face the following two major challenges: i) how to learn unified representations that are aligned for stock pools with different sizes and stocks; ii) how to guide RL agents to construct portfolios that concentrate on most favorable stocks and neglect the stocks outside the target pool. To tackle these two challenges, we have designed an RL framework that called EarnMore with one-time training that takes into account the distinct investment preferences of each investor and invests in various CSPs. This framework allows for dynamic adjustments to the pool in the investment process, contributing to a more tailored and effective portfolio. Our contributions are four-fold:

- We introduce a learnable masked token to represent unfavorable stocks, which enables the unified representation of stock pools with different sizes and stocks.
- We derive meaningful embeddings using a self-supervised masking and reconstruction process that captures stock relationships.
- We propose a re-weighting mechanism to rescale the distribution of portfolios to make it concentrate on favorable stocks and neglect stocks outside the target pool.
- Experiments on 8 subset stock pools of the US stock market demonstrate the superiority of EarnMore over 14 baselines in terms of 6 popular financial metrics with one-time training.

## 2 RELATED WORK

### 2.1 Portfolio Management

Portfolio management is an essential aspect of investment and involves a strategic allocation of resources to achieve optimal returns and avoid risks simultaneously. There are two commonly used traditional rule-based methods, i.e., mean reversion [22] and momentum [13]. The former buys low-priced stocks and sells high-priced ones, whereas the latter relies on recent performance with the expectation that trends will continue. Cross-Sectional Momentum [14] and Time-Series Momentum [20] are two classical momentum trading methods. However, traditional rule-based methods are difficult to capture fleeting patterns in changing market conditions and perform well only in specific scenarios [4].

In the past few years, advanced prediction-based methods have significantly surpassed traditional rule-based methods in performance. These methods treat PM as a supervised learning task and predict future returns (regression) or price movements (classification). Then, the heuristic strategy generator allocates asset investments based on the prediction results [34] [2]. Specifically, prediction-based methods can be categorized into two kinds, machine learning models like XGBoost [3] and LightGBM [16], along with deep learning models such as ALSTM [23] and TCN [1]. However, the volatility and noisy nature of the financial market makes it extremely

difficult to accurately predict future prices [7]. Furthermore, the gap between prediction signals and profitable trading actions [10] is difficult to bridge. Therefore, prediction-based methods do not perform satisfactorily in general.

Recent years have witnessed the successful marriage of reinforcement learning and portfolio management due to its ability to handle sequential decision-making problems. EIIE [15] utilizes the convolutional neural network for feature extraction and RL for decision-making. The Investor-Imitator [5] framework demonstrates its utility in financial investment research by emulating investor actions. SARL [33] leverages the price movement prediction as additional states based on deterministic policy gradient methods. DeepTrader [30] dynamically balances risk-return with market indicators and utilizes a unique graph structure to generate portfolios. HRPM [29] presents a hierarchical framework that addresses long-term profits and considers price slippage as part of the trading cost. DeepScalper [27] combines intraday trading and risk-aware tasks to capture the investment opportunities [3].

### 2.2 Masked Autoencoders

Masked Autoencoders (MAEs) [12] are neural networks employed in self-supervised learning to obtain effective embeddings. An autoencoder is designed to learn a representation for a set of data, which is initially used for self-supervised learning in image, video and audio. Recently, they have been extensively studied in the field of time series prediction. PatchTST[21] enhances multivariate time series forecasting through self-self-supervised learning. It achieves this by partitioning time series data into patches and utilizing separate channels for univariate time series. This approach boosts memory efficiency and improves the model's ability to capture historical patterns. Several notable works have used MAEs to learn time series representations, such as SimMTM [6] and Ti-MAE [18].

In financial markets, which tend to have lower signal-to-noise ratios than typical time series data. Leveraging MAEs for self-supervised learning, we efficiently reduce data dimensionality, filter noise, and highlight essential information. MAEs uncover hidden stock relationships and represent investor-unfavorable stocks with masked tokens, improving investor-agent interactions in PM.

## 3 PRELIMINARIES

In this section, we present definitions and formulas for necessary terms in PM. Next, we provide a Markov Decision Process (MDP) model for portfolio management with CSPs.

### 3.1 Definitions and Formulas

DEFINITION 1. *(OHLCV). Open-High-Low-Close-Volume is a type of bar chart obtained from the financial market. The OHLCV of stock $i$ at time $t$ is denoted as $\boldsymbol{P}_{i,t} = [p_{i,t}^o, p_{i,t}^h, p_{i,t}^l, p_{i,t}^c, v_{i,t}]$, where $p_{i,t}^o$, $p_{i,t}^h$, $p_{i,t}^l$, $p_{i,t}^c$ and $v_{i,t}$ are open, high, low, close prices and volume.*

DEFINITION 2. *(Technical indicators). A technical indicator indicates a feature calculated by a formulaic combination of the historical OHLCV. We denote the technical indicator vector at time $t$: $\boldsymbol{Y}_t = [y_t^1, y_t^2, ..., y_t^K]^T \in \mathbb{R}^K$. For $k \in [1, K]$, $y_t^k$ is represented as*

---

[2]For instance, top-k [34] assets are chosen for investment allocation, with the proportion determined by the forecasted future returns ranked by magnitude.

[3]HRPM and Deepscalper are excluded from the experimental section due to the limited order book being introduced as extra data in addition to k-line prices.

$y_t^k = f_t^k(x_{t-D+1}, ..., x_{t-1}, x_t|\theta^k)$, where $D$ denotes past time steps up to $t$, and $\theta^k$ are hyperparameters for indicator $k$.

DEFINITION 3. *(Portfolio). A portfolio is a combination of financial assets, denoted as $\mathbf{W}_t = [w_t^0, w_t^1, ..., w_t^N] \in \mathbb{R}^{N+1}$, with $N + 1$ assets, including 1 risk-free cash and $N$ risky stocks. Each asset $i$ is assigned a weight $w_t^i$ representing its portfolio proportion, subject to the constraint that $\sum_{i=0}^N w_t^i = 1$ for full investment.*

DEFINITION 4. *(Portfolio Value). Portfolio value at time step $t$, denoted as $V_t$, represents the sum of individual asset values in the portfolio, $V_0$ represents the initial cash and $V_t$ calculated using stock closing prices through the following formula:*

$$V_t = w_t^0 V_{t-1} + (1 - w_t^0) V_{t-1} (1 + \sum_{i=1}^N w_t^i \frac{p_{i,t}^c - p_{i,t-1}^c}{p_{i,t-1}^c}). \tag{1}$$

## 3.2 Problem Formulation

We model portfolio management as a Markov Decision Process (MDP) and provide a detailed description of the MDP modeling process for portfolio management with CSPs in this section.

**MDP Formulation for PM.** We formulate PM as an MDP following a standard RL scenario, where an agent (investor) interacts with an environment (the financial markets) in discrete time to make actions (investment decisions) and get rewards (profits). In this work, the objective is to maximize the final portfolio value within a long-term investment time horizon. We formulate PM as an MDP, which is constructed by a 5-tuple $(S, \mathcal{A}, T, R, \gamma)$. Specifically, $S$ is a finite set of states. $\mathcal{A}$ is a finite set of actions. The state transition function $T : S \times \mathcal{A} \times S \rightarrow [0, 1]$ encapsulates transition probabilities between states based on chosen actions. The reward function $R : S \times \mathcal{A} \rightarrow R$ quantifies the immediate reward of taking an action in a state. The discount factor is $\gamma \in [0, 1)$. A policy $\pi : S \times \mathcal{A} \rightarrow [0, 1]$ assigns each state $s \in S$ a distribution over actions, where $a \in \mathcal{A}$ has probability $\pi(a|s)$.

**MDP Formulation for PM with CSPs.** Existing work [30, 33] focus on the GSP and lacks formal modeling in the broader context of CSPs. We denote a GSP as $U$, consisting of $N$ individual stocks. By randomly masking some stocks that investors are unfavorable to, it can generate a diverse set of CSPs, which are subsets of $U$. We define a sub-pool of GSP $U$ that masks $N^*$ stocks at time step $t$ to create CSP $C_t$, where $|C_t| = N - N^*$. We model an MDP for PM with CSP $C_t$ using maskable stock representation[4]. The widely used MDP formulation in existing work is a special case in our PM with CSPs formulation when $C_t = U$. The details of the PM with CSPs as an MDP are set as follows:

- **State**. The GSP stocks time series are composed of three components: historical OHLC prices $\mathbf{P}_t$, classical technical indicators $\mathbf{Y}_t$, and temporal information $\mathbf{D}_t$. We denote the feature of GSP $U$ during the historical $D$ time steps as $\mathbf{X}_t = [\mathbf{P}_t, \mathbf{Y}_t, \mathbf{D}_t] = [x_{t-D+1}, x_{t-D+2}, ..., x_t] \in \mathbb{R}^{N \times D \times F}$, $F$ represents the feature dimension. After masked $N^*$ stocks, the feature of CSP $C_t$ is denoted as $\mathbf{X}_t^* = [x_t^1, x_t^2, ..., x_t^{N-N^*}] \in \mathbb{R}^{(N-N^*) \times D \times F}$. To restore the GSP feature dimension, the process entails duplicating and populating with a learnable masked token $[M]$. Then, the state is represented as $s_t = [x_t^1, x_t^2, ..., x_t^{N-N^*}, [M], [M], ..., [M]] \in \mathbb{R}^{N \times D \times F}$, where the number of $[M]$ is $N^*$.

- **Action.** Given the state, the action of the agent at time step $t$ can be entirely represented by the portfolio vector $a_t = \mathbf{W}_t = [w_t^0; w_t^1, ..., w_t^{N-N^*}; w_{[M],t}^1, w_{[M],t}^2, ..., w_{[M],t}^{N^*}] \in \mathbb{R}^{N+1}$. The proportion of cash retained is represented by $w_t^0$, $[w_t^1, ..., w_t^{N-N^*}]$ denotes the proportion of $N - N^*$ investor favorable stocks, $[w_{[M],t}^1, w_{[M],t}^2, ..., w_{[M],t}^{N^*}]$ denotes the proportion of $N^*$ investor unfavorable stocks.

- **Reward**. Drawing on prior research [19], for each trading time step $t$, we equate the reward $r_t$ with the portfolio value $V_t$.

## 4 EARNMORE

As shown in Figure 2, we present a reinforcement learning framework called EarnMore with one-time training on a GSP, enabling invest on various CSPs and achieving optimal investment portfolios. With this framework, investors have the flexibility to invest in stock pools aligned with their individual preferences. Furthermore, during the trading process, these stock pools can be dynamically adapted to construct more efficient and tailored portfolios, aligning with investors' current decisions. EarnMore consists of three main components: i) a unified approach for representing customized stock pools with different sizes and stocks, which we call maskable stock representation (**§4.1**); ii) reinforcement learning optimization procedures for PM with CSPs (**§4.2**); iii) a re-weighting mechanism that concentrates on favorable stocks and neglects unfavorable ones to rescale the distribution of portfolios (**§4.3**).

## 4.1 Maskable Stock Representation for CSPs

Consistent representation is crucial for CSPs with different sizes and stocks before portfolio decisions. If only the stocks within the CSP are embedded into the agent for decision-making, and discarding the masked-out stocks will lead to three issues. Firstly, the action dimension of agent cannot adapt to different CSP sizes. Secondly, the agent may perform poorly or even fail due to the inability to distinguish between the different CSPs when dealing with CSPs of the same size but different stocks. Finally, discarding unfavorable stocks may lead to a loss of relationships between stocks, which will negatively impact performance. To address them, we introduce a maskable stock representation that identifies the position of each stock within the GSP. We employ two levels of stock representation, which are stock-level in Module $(a)$ and pool-level constructed through masking and reconstruction in Module $(b)$. This process reveals hidden connections between stocks, and our maskable stock representation is based on the pool-level.

**Learning for Stock-level Representation.** In Equation 2, we exploit stock features (prices and technical indicators) and temporal characteristics for stock-level representation. Following the approach in Timesnet [31], we employ 1D convolution to produce dense embeddings for stock features and utilize an embedding layer to handle sparse temporal features. The final stock-level representation is formed by the summation of these dense and sparse embeddings. It is defined as follows:

$$l_s(\mathbf{X}_t) = \psi_e(\mathbf{D_t}; \theta_e) + \psi_c(\mathbf{P}_t, \mathbf{Y}_t; \theta_c), \tag{2}$$

where $\psi_e, \psi_c$ denote the embedding layer and 1D convolutional layer, and $\theta_e, \theta_c$ are their learnable parameters respectively.

---

[4]Detailed description of the maskable stock representation can be found in section 4.1.

**Figure 2: The overall architecture of EarnMore. Module (a) is used to extract stock-level embeddings from GSP. Module (b) is the masking and reconstruction process to learn pool-level embeddings. Module (c) is an agent with masked token awareness.**

**Learning for Pool-level Representation.** Stock-level representation describes the vertical time series information within each individual stock without capturing the horizontal inter-stock representation. In our PM with CSPs environment, directly masking certain stocks could potentially result in losing important and valuable connections between stocks when using them as stock representations. To address this limitation, we introduce the pool-level representation, which strengthens the connections between stocks in the GSP through the masking and reconstruction process. Notably, we employ stock-level embedding as the local embedding to replace the patching embedding for historical data employed in MAEs [12] or PatchTST [21].

During training phase, we utilize the adaptive masking strategy developed by MAGE [17] to simulate various CSPs with different numbers and compositions of stocks, which helps to improve the representational capability of the pool-level embedding. We sample a masking ratio $r$ from a truncated Gaussian distribution:

$$g(r; \mu, \sigma, a, b) = \varphi(\frac{r - \mu}{\sigma}) / (\Phi(\frac{b - \mu}{\sigma}) - \Phi(\frac{a - \mu}{\sigma})), \quad (3)$$

where $\varphi(\cdot)$ is the probability density function of the standard normal distribution, $\Phi(\cdot)$ is its cumulative distribution function, $a$ and $b$ are the lower and upper bounds.

In Equation 4, the process for constructing maskable stock representation is outlined through an encoder and decoder procedure. The process starts with the encoder phase, a random masking ratio $r$ is sampled and then is used to mask a subset of stock-level embeddings selectively using the masking operation $\eta_{mo}$. Only the unmasked embeddings are retained and subsequently fed into the encoder $\psi_{enc}$ to extract latent embeddings. During the decoder phase, the latent embeddings are filled to the number of stock-level embeddings using a learnable masked token called $m$ via the mask-filled operation $\eta_{mf}$. Finally, the decoder $\psi_{dec}$ is used to reconstruct the price $\tilde{\mathbf{P}}_t$ of masked stocks:

$$\begin{aligned} l_p(\mathbf{X}_t) &= \psi_{enc}(\eta_{mo}(l_s(\mathbf{X}_t), g(r; \mu, \sigma, a, b)); \theta_{enc}) \\ \rho(\mathbf{X}_t, m) &= \eta_{mf}(l_p(\mathbf{X}_t), m) \\ \tilde{\mathbf{P}}_t &= \psi_{dec}(\rho(\mathbf{X}_t, m); \theta_{dec}), \end{aligned} \quad (4)$$

where $\psi_{enc}, \psi_{dec}$ denote the Encoder and Decoder, $\theta_{enc}, \theta_{dec}$ are their respective learnable parameters.

After filling in the masked token, we refer to the resulting latent embedding as maskable stock representation, and we abbreviate $\eta_{mf}(l_p(\mathbf{X}_t), m)$ as $\rho(\mathbf{X}_t, m)$, which will be used as state for portfolio decision making in the reinforcement learning process. The masked token makes the agent sense which stocks are unfavorable to the investor, thus catering to the investor's preferences and personal decisions, and enabling collaboration between the agent and investor, who has access to various sources of information and have some expectation of the future direction of specific stocks. It is important to mention that we retain the [CLS] token to understand how cash is allocated. This token can capture the overall sequence representation and preserve global sequence information when decoding, which is exactly what we need.

## 4.2 RL Optimization for PM with CSPs

Our reinforcement learning training process is based on the Soft Actor-Critic (SAC) [11]. There are two main components called Actor and Critic in RL optimization. The Actor utilizes the latent embeddings populated by masked tokens to generate actions that indicate the allocation ratios for cash and individual stocks, and the Actor is aware of masked token for unfavorable stocks and will avoid allocating them during decision-making. The Critic evaluates portfolio performance using populated latent embeddings with masked tokens and actions that the Actor generates. This evaluation provides a scoring mechanism that guides the learning process and helps optimize the portfolio management strategy.

**Optimization for Q-Value Network.** We use maskable stock representation $\rho(\mathbf{X}_t, m)$ defined in Equation 4 instead of raw market data as input states $\rho$ for Actor and Critic. Let $Q_\theta(s, a) = Q_\theta(\rho, a)$ represent the $Q$-value function, and $\pi_\phi(\rho, a)$ denote the policy function. Assumed that the output of $\pi_\phi$ follows a normal distribution with expectation and variance, the $Q$-value function can be learned by minimizing the flexible Bellman residuals:

$$\begin{aligned} J_Q(\theta) &= \mathbb{E}_{(s_t, a_t) \sim \mathcal{D}} [\frac{1}{2} (Q_\theta(\rho_t, a_t) - (r(s_t, a_t) + \gamma \mathbb{E}_{s_{t+1} \sim p} [V_{\bar{\theta}}(\rho_{t+1})]))^2] \\ V_{\bar{\theta}}(\rho_t) &= \mathbb{E}_{a_t \sim \pi} [Q_{\bar{\theta}}(\rho(s_t, m), a_t) - \alpha \log \pi_\phi(a_t | \rho_t)], \end{aligned} \quad (5)$$

where $\rho(s_t, m)$ abbreviated as $\rho_t$, $Q_{\bar{\theta}}$ represents the target Q-value network, $\bar{\theta}$ is the exponential moving average of the parameter $\theta$.

**Optimization for Policy Network.** To optimize $J_\pi(\phi)$, we utilize the reparameterization technique for the policy network $\pi_\phi$. This technique involves representing $\pi_\phi$ as a function that takes the state $s$ and standard Gaussian samples $\epsilon$ as inputs and directly outputs the action $a = f_\phi(\epsilon; s)$. Assuming $\mathcal{N}$ is the standard normal distribution, $\pi_\phi$ can be derived by minimizing KL divergence:

$$J_\pi(\phi) = \mathbb{E}_{s_t \sim D, \epsilon_t \sim \mathcal{N}}[\alpha \log \pi_\phi(f_\phi(\epsilon_t; \rho_t)|s_t) - Q_\theta(\rho_t, f_\phi(\epsilon_t; \rho_t))]. \quad (6)$$

**Optimization for Parameter Alpha.** We employ an automatic entropy tuning method to adjust parameter $\alpha$ by minimizing the following loss function:

$$J(\alpha) = E_{a_t \sim \pi_t}[-\alpha \log \pi_t(a_t|\rho(s_t, m)) - \alpha\bar{\mathcal{H}}], \quad (7)$$

where $\bar{\mathcal{H}}$ is the target entropy hyperparameter.

**Optimization for Maskable Stock Representation.** In the masking and reconstruction process, we optimize the maskable stock representation using mean-squared error. Reconstruction losses are calculated based only on the price of masked stocks:

$$J(\theta_e, \theta_c, \theta_{enc}, \theta_{dec}) = \frac{1}{N^*} \sum_{i=1}^{N^*} (p_{i,t} - \tilde{p}_{i,t})^2, \quad (8)$$

where $N^*$ represents the number of masked stocks.

Our implementation process follows the same optimization process for each batch of SAC. We first optimize the Q-value network, followed by the alpha, strategy network, and maskable stock representation. However, we find that using weighted sum loss to optimize both the maskable stock representation and the remaining three components will have a negative impact on the distribution of data sampled by the RL process and lead to unstable training.

### 4.3 Re-weighting Method

In a continuous decision space in portfolio management, agents face difficulties in making accurate decisions. For instance, in a constantly changing market environment, agents may overfit a fixed number of market patterns and struggle to react quickly in high-volatility markets. These issues can lead to agents micro-investing in stocks with low future returns or even result in losses.

In our setting for PM with CSPs, we encounter unique problems in addition to those already present in PM: i) the state that we input to the agent is latent embeddings containing filled masked tokens, the agent may be investing in the masked stocks that these investors expect to lose money, which is precisely what investors do not want to witness; ii) due to the extent of error in decision-making, part of the investment proportion of high-yield stocks may be taken by stocks with low or negative expected future returns.

Both problems can be solved by portfolio sparsification. Drawing inspiration from the Boltzmann distribution and Gumbel-Softmax, we introduce an additional hyperparameter $T$ to the softmax function for re-weighting portfolios to achieve sparsification of tiny investment proportions to zero:

$$Re(\mathbf{x}) = e^{x_i/T} / \sum_{j=1}^{N} e^{x_j/T}. \quad (9)$$

In this context, $\mathbf{x}$ represents the Actor's logits for portfolio, and $T \in (0, \infty)$ is a temperature parameter. Lower $T$ values lead to sparser allocations. As $T$ approaches 0, all investments tend to allocate to the asset with the highest expected return. For $T = 1$, re-weighting degenerates to softmax, while for $T > 1$, it reduces the allocation variance and even leads to equal allocation.

## 5 EXPERIMENTS

In this section, we conduct a series of experiments to evaluate the proposed framework. First, we demonstrate that our approach achieves better returns in two real US financial markets and substantially outperforms the baseline methods in the global stock pool. Next, we construct 6 customizable stock pools based on 3 different investor investment preferences in two US financial markets to demonstrate that our framework can effectively meet investors' preferences and decisions in the trading process. Finally, we conduct ablation studies to answer the following questions:
**RQ1**: How is the usefulness of each component of EarnMore?
**RQ2**: Why are direct methods for PM with CSPs not working?
**RQ3**: How is the efficiency of the EarnMore model?

### 5.1 Datasets and Processing

In our experiments, we used daily k-line data for 10,273 US stocks from Yahoo Finance. We derived technical indicators from OHLCV data to understand market trends, volatility, and liquidity. After preprocessing to address data quality issues, we ended up with 3,094 US stocks and 95 technical indicators based on Qlib's Alpha158 [32]. Our analysis covered the period from September 2007 to June 2022, which included significant events such as the subprime mortgage crisis, the COVID-19 pandemic, and geopolitical conflicts.

To demonstrate the generality of EarnMore, we purposely chose two stock indices, SP500 and DJ30, with very different sizes. The SP500 consists of 420 stocks and the DJ30 consists of 28 stocks, where we exclude some stocks with many missing values. According to the Global Industry Classification Standard[5] (GICS), we categorize SP500 and DJ30 into 49 and 24 industries at the industry level. Examples of industries include banking, insurance, software services, automotive manufacturing, and so on.

**Table 1: Datasets and Date Splits for SP500 and DJ30 Indices**

| Dataset | SP500 | | DJ30 | |
|---|---|---|---|---|
| | Stock | Industry | Stock | Industry |
| GSP | 420 | 49 | 28 | 24 |
| CSP1 | 62 | 8 | 10 | 9 |
| CSP2 | 39 | 7 | 7 | 6 |
| CSP3 | 168 | 28 | 10 | 8 |
| | Train | | Test | |
| Date Split | 2007-09-26 ~ 2018-01-25 | | 2018-01-26 ~ 2019-07-22 | |
| | 2007-09-26 ~ 2019-07-22 | | 2019-07-23 ~ 2021-01-08 | |
| | 2007-09-26 ~ 2021-01-07 | | 2021-01-07 ~ 2022-06-26 | |

We conduct a comprehensive evaluation to validate our framework's effectiveness and performance in various real-world scenarios. We consider two main factors in the evaluation: i) events and markets under different conditions, e.g. COVID-19, geopolitical conflicts, bull and bear; ii) customizable stock pools with different investors' investment preferences, such as one investor prefers investing in the technology and communication industries, and another one prefers investing in financial and insurance industries. Based on the above statements as shown in Table 1, we choose 3 date periods as test datasets from 2018-01-16 to 2022-06-26, containing events such as COVID-19 and other complex market conditions. We

[5]https://www.msci.com/our-solutions/indexes/gics

carefully select three CSPs based on three different investor preferences, with CSP1, CSP2, and CSP3 corresponding to the technology, financial, and service as the main industries. To better reflect the real-world demand for industry diversity in PM, we randomly add several stocks from other industries into the three CSPs.

## 5.2 Evaluation Metrics

We compare EarnMore and baselines in terms of 6 financial metrics, including 1 profit criterion, 3 risk-adjusted profit criteria, and 2 risk criteria. Definitions and formulas are available in Appendix A.1.

## 5.3 Baselines

To provide a comprehensive comparison of EarnMore, we select 14 state-of-the-art and representative stock prediction methods of 4 different types consisting of 3 rule-based (**Rule-based**) methods, 2 machine learning-based (**ML-based**) methods, 2 deep learning-based (**DL-based**) methods and 7 reinforcement learning-based (**RL-based**) methods. Descriptions of baselines are as follows.

- **Rule-based Methods**: **BLSW** [22] is based on mean reversion that buys underperforming stocks and sells outperforming ones. **CSM** [14] is a momentum strategy that prefers assets with recently strong performance and expects short-term success.
- **ML-based Methods**: **XGBoost** [3] leverages Gradient Boosting Decision Tree (GBDT) for accurate predictions in supervised learning tasks. **LightGBM** [16] is an efficient GBDT with gradient-based one-side sampling and exclusive feature bundling.
- **DL-based Methods**: **ALSTM** [23] ALSTM is a Recurrent Neural Network (RNN) that uses an external attention layer to gather information from all hidden states. **TCN** [1] is a Convolutional Neural Network (CNN) architecture for sequence modeling in time series analysis and natural language processing.
- **RL-based Methods**: **PG** [28] optimizes policy function while considering risk and market conditions in PM without estimating value function. **SAC** [11] is an off-policy actor-critic algorithm that optimizes investment strategies in PM using entropy regularization and soft value functions in continuous portfolio action spaces. **PPO** [24] updates investment policies iteratively to balance exploration and exploitation, ensuring stability and sample efficiency in PM. **EIIE** [15] is the first work formulating the PM problem as an MDP, and it outperforms traditional PM methods by using CNN for feature extraction and RL for portfolio decisions. **SARL** [33] proposes a state-augmented RL framework, which leverages the price movement prediction as additional states based on deterministic policy gradient methods. **Investor-Imitator (IMIT)** [5] shows good performance as a RL-based framework in PM by replicating investor actions. **DeepTrader** [30] leverages RL to model inter-stock relationships and balance the risk-return trade-offs.

## 5.4 Implement Details

For the *encoder*, *decoder*, *actor*, and *critic*, each of them consists of 2 layers of Multi-Layer Perceptron (MLP) with GELU activation function. We set the horizon length to 128, batch size to 128, and buffer size to $1e5$. The embedding dimension is 64 for all components. There are three AdamW optimizers, an optimizer for maskable stock representation learning and two for the actor and critic in RL optimization. Mean Squared Error (MSE) is used for the learning and optimization process.

Experiments are conducted on an Nvidia A6000 GPU, and we use grid search to determine the hyperparameters. The multi-step learning rate scheduler with warm-up starts at $1e-8$, increases to $1e-5$ after 300 episodes, and multiplied by 0.1 at the 600th, 1000th, and 1400th episodes. Masking method parameters are $a = 0.6$, $b = 0.8$, $\mu = 0.7$, and $\sigma = 0.1$. The optimal temperature parameter $T$ is found to be 0.1. The implementation of those ML-based and DL-based methods is based on Qlib[32]. As for other baselines, we use the default settings in their public implementations. We run experiments with individual 9 runs using 3 date splits × 3 random selected seeds and report the average performance[6].

## 5.5 Results and Analysis

The performances of EarnMore and other baseline methods in GSPs on the two US financial markets SP500 and DJ30 are shown in Table 2. Specifically, the results of cumulative return are drawn in Figure 3. We also train and test each baseline method individually for each CSP, for the method shows no adaptive ability in customized stock pools. We then compared the 3 state-of-the-art RL-based methods with EarnMore on two important metrics, as shown in Table 3. Furthermore, to evaluate EarnMore's ability to adapt to investors' personal decisions during trading process, we point out several real-world scenarios of adjusting CSP and display the dynamic changes of EarnMore in Figure 4.

**Performance on Global Stock Pools.** We compared EarnMore with 14 baseline methods in terms of 6 financial metrics. Table 2 and Figure 3 demonstrate our framework outperforms others on portfolio management with higher returns in GSPs. For the SP500, EarnMore achieves the highest ARR of 97% and SR of 2.032, significantly higher than the second-best method. For the DJ30, EarnMore achieves improvements in terms of ARR, SR, CR, and SoR by 46.7%, 8.9%, 6.2%, and 2.4%. We can also observe that ML-based methods are optimal in controlling risk, but not outstanding in capturing returns. The reason behind it is that tree models are more robust to outliers and noise in the data, and thus can adaptively capture non-linear relationships to reduce decision risk. Specifically, ALSTM achieved a surprising 43.5% return on the SP500, due to large returns from several decisions in a large number of bad decisions, and thus we do not recommend using it. Besides, it is worth noting that the higher the potential return, the higher the risk involved in portfolio management. EarnMore is slightly inferior yet comparable to baseline methods on risk metrics, i.e., MDD and VOL. As for DJ30 dataset, EarnMore fails to perform well in MDD but achieves over 40% improvement in ARR to DeepTrader, which is the second best overall. Thus, for EarnMore, it is a slight compromise on risk control, as our priority is to maximize final portfolio values.

The global COVID-19 pandemic reached its peak between February 14 and March 20 2020, causing a significant decline in the economy and intense investor concerns. This led to a substantial fall in the stock market, with the SP500 and DJ30 indices falling by 31.81% and 34.78%, respectively. As shown in Figure 3, EarnMore is far less affected in returns than baseline methods and continues to gain returns after the market rebounded. Even during market

---

[6]We report the experiment results of EarnMore if not particularly pointed out.

**Table 2: Performance comparison on SP500 and DJ30 with Global Stock Pool. Results in red, yellow and green show the best, second best and third best results on each dataset.**

| Stock Index | | SP500 | | | | | | DJ30 | | | | |
|---|---|---|---|---|---|---|---|---|---|---|---|---|
| | | Profit | Risk-Adjusted Profit | | | Risk Metrics | | Profit | Risk-Adjusted Profit | | | Risk Metrics | |
| Categories | Strategies | ARR%↑ | SR↑ | CR↑ | SOR↑ | MDD%↓ | VOL↓ | ARR%↑ | SR↑ | CR↑ | SOR↑ | MDD%↓ | VOL↓ |
| Rule-based | Market | 9.320 | 0.556 | 0.702 | 17.120 | 26.160 | 0.014 | 6.710 | 0.458 | 0.776 | 15.560 | 22.200 | 0.013 |
| | BLSW | 11.630 | 0.696 | 0.894 | 21.450 | 24.560 | 0.013 | 7.610 | 0.512 | 0.857 | 16.930 | 21.540 | 0.012 |
| | CSM | 5.070 | 0.329 | 0.434 | 9.840 | 23.350 | 0.013 | 5.930 | 0.400 | 0.643 | 12.950 | 20.770 | 0.012 |
| ML-based | XGBoost | 10.690 | 0.377 | 0.473 | 13.650 | 19.300 | 0.016 | 10.260 | 0.343 | 0.599 | 10.420 | 14.760 | 0.013 |
| | LightGBM | 16.330 | 0.575 | 0.744 | 20.110 | 24.760 | 0.016 | 13.420 | 0.591 | 0.703 | 14.220 | 20.900 | 0.014 |
| DL-based | ALSTM | 43.50 | 1.157 | 1.367 | 22.501 | 35.820 | 0.026 | 15.030 | 1.186 | 0.590 | 14.890 | 28.070 | 0.013 |
| | TCN | 13.560 | 1.044 | 1.460 | 14.540 | 35.780 | 0.025 | 6.980 | 0.732 | 0.269 | 8.280 | 37.400 | 0.018 |
| RL-based | PG | 12.580 | 0.431 | 0.519 | 24.340 | 26.180 | 0.014 | 7.970 | 0.321 | 0.435 | 8.430 | 21.570 | 0.012 |
| | PPO | 15.130 | 0.537 | 0.742 | 14.770 | 24.100 | 0.013 | 9.240 | 0.385 | 0.512 | 10.140 | 20.810 | 0.012 |
| | SAC | 15.140 | 0.538 | 0.743 | 14.770 | 24.100 | 0.013 | 9.150 | 0.326 | 0.448 | 8.830 | 20.600 | 0.012 |
| | EIIE | 15.030 | 0.540 | 0.627 | 15.450 | 26.920 | 0.015 | 22.900 | 0.689 | 1.465 | 23.450 | 16.770 | 0.014 |
| | SARL | 21.240 | 0.756 | 0.970 | 21.230 | 24.000 | 0.013 | 21.920 | 0.786 | 1.109 | 23.020 | 20.400 | 0.012 |
| | IMIT | 50.300 | 1.162 | 1.949 | 35.050 | 25.420 | 0.018 | 27.640 | 0.909 | 1.593 | 27.380 | 20.050 | 0.014 |
| | DeepTrader | 60.290 | 1.980 | 2.195 | 34.260 | 28.580 | 0.013 | 32.230 | 1.335 | 1.440 | 27.110 | 21.190 | 0.013 |
| | **EarnMore** | 97.170 | 2.032 | 2.506 | 42.160 | 28.120 | 0.023 | 47.290 | 1.454 | 1.692 | 28.040 | 21.650 | 0.018 |
| Improvement over SOTA | | 61.171% | 2.626% | 14.169% | 20.285% | - | - | 46.727% | 8.914% | 6.215% | 2.411% | - | - |

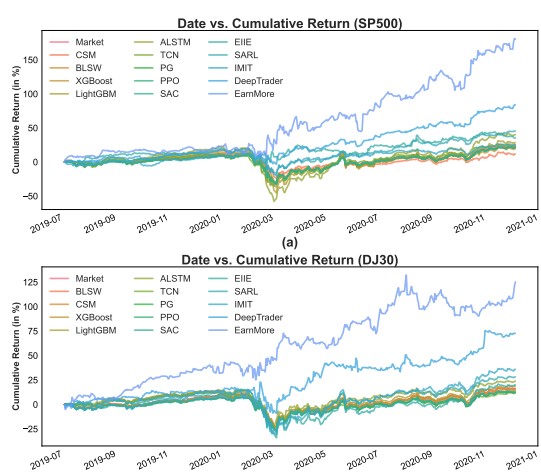

Figure 3: Performance on GSP for SP500 and DJ30

**Table 3: Performance comparison on SP500 and DJ30 with Customizable Stock Pools. Underlined metrics indicate the best-performing results.**

| Stock Index | | SP500 | | DJ30 | |
|---|---|---|---|---|---|
| Pool | Strategies | ARR%↑ | SR↑ | ARR%↑ | SR↑ |
| CSP1 | SARL | 34.330 | 0.820 | 24.140 | 0.638 |
| | IMIT | 20.973 | 0.860 | 20.071 | 0.920 |
| | DeepTrader | 34.030 | 0.793 | 27.740 | 0.757 |
| | **EarnMore** | 122.610 | 2.278 | 53.990 | 1.810 |
| CSP2 | SARL | 17.000 | 0.570 | 20.020 | 0.820 |
| | IMIT | 7.971 | 0.486 | 11.841 | 0.751 |
| | DeepTrader | 34.030 | 0.793 | 38.470 | 0.955 |
| | **EarnMore** | 110.110 | 2.279 | 43.400 | 1.549 |
| CSP3 | SARL | 18.090 | 0.760 | 10.910 | 0.480 |
| | IMIT | 21.193 | 1.220 | 6.851 | 0.496 |
| | DeepTrader | 61.320 | 1.489 | 16.840 | 0.601 |
| | **EarnMore** | 93.670 | 2.120 | 43.460 | 1.572 |

downturns, EarnMore is able to identify stocks with the potential to generate higher returns when the market rebounds.

**Performance on Customizable Stock Pools.** We illustrate the effectiveness of CSPs in two aspects. Firstly, we compare the profitability performance of CSPs - formulated according to investor preferences - with 3 state-of-the-art RL-based methods. Secondly, we demonstrate the adaptability and robustness of EarnMore to investors' personal decisions in the trading process.

EarnMore achieves impressive performances on all CSPs, as shown in Table 3. Specifically, in the CSP1 stock pool, which consists of technology stocks, EarnMore's profitability is significantly higher compared to other methods. It is consistent with the notable increasing values of technology stocks over an extended period of time, and thus demonstrates that our method provides more scope for profit-seeking. In the other two CSPs, which are in the financials and services industries, EarnMore also delivers notable

return improvements. Overall, our method is able to automatically adapt to investors' preferences and generate substantial returns.

General Electric Company (GE) was delisted from the SP500 on June 26, 2018. As shown in Figure 4(a), after removing GE from the stock pool of the SP500 on the date of June 26, 2018, EarnMore can adapt itself to investor decisions to achieve a small return increase. Between March 2022 and June 2022, the stock price of Apple (APPL) dropped sharply by nearly 25% due to several factors, including the impact of the war and Apple's mobile phone downtime incident. As shown in Figure 4(b), excluding AAPL from DJ30 can significantly improve returns. It is important to mention that we add Microsoft (MSFT) technology stock to the CSP2 of the SP500. We purposely chose periods when the MSFT price was decreasing, to test the strength and robustness of EarnMore, in case an investor inadvertently selects an unsuitable stock. As depicted in

Figure 4(c), EarnMore is able to decrease its MSFT investment by properly screening the stock with minimal to no impact on the overall returns throughout the trading procedure. Goldman Sachs (GS) announced the advancement of steel project and new energy vehicle development, which released a signal that stock price of GS would rise in 2021. Thus, our model adds GS to the stock pool, as shown in Figure 4(d), and gets more significant return growth.

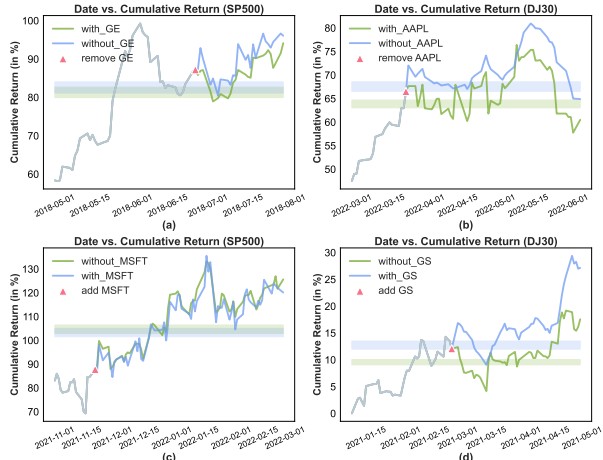

Figure 4: Performance on CSPs with dynamic changes

## 5.6 Ablation Study

**Table 4: Ablation Study of EarnMore. Red, green and underline indicate improvement, performance decrease and best results, respectively.**

| Stock Index | | SP500 | | | | | | DJ30 | | | | |
|---|---|---|---|---|---|---|---|---|---|---|---|---|
| Pool | Model | ARR | $\Delta_{ARR}$ | SR | $\Delta_{SR}$ | MDD | $\Delta_{MDD}$ | ARR | $\Delta_{ARR}$ | SR | $\Delta_{SR}$ | MDD | $\Delta_{MDD}$ |
| GSP | w/o-MR | 33.2 | - | 1.49 | - | 22.7 | - | 10.5 | - | 0.71 | - | 21.5 | - |
| | w/o-M | 74.8 | 125.5 | 1.89 | 27.3 | 22.5 | -1.2 | 31.4 | 198.6 | 1.14 | 60.8 | 19.4 | -9.5 |
| | EarnMore | 97.2 | 29.9 | 2.03 | 7.23 | 28.1 | 25.3 | 47.3 | 50.7 | 1.45 | 28.1 | 21.7 | 11.4 |
| CSP1 | w/o-MR | 32.6 | - | 1.18 | - | 25.4 | - | 14.2 | - | 0.78 | - | 20.5 | - |
| | w/o-M | 46.7 | 43.3 | 1.09 | -7.6 | 32.2 | 26.5 | 23.9 | 69.5 | 1.02 | 30.9 | 22.2 | 8.50 |
| | EarnMore | 122.6 | 162.3 | 2.28 | 108.4 | 25.1 | -22.1 | 53.9 | 125.0 | 1.81 | 78.2 | 22.9 | 3.09 |
| CSP2 | w/o-MR | 8.46 | - | 0.62 | - | 28.2 | - | 11.02 | - | 0.81 | - | 20.1 | - |
| | w/o-M | 18.14 | 114.4 | 0.70 | 13.9 | 32.1 | 13.9 | 22.2 | 8.50 | 0.02 | 31.7 | 21.7 | 8.26 |
| | EarnMore | 110.1 | 507.0 | 2.28 | 223.7 | 25.8 | -19.7 | 43.4 | 95.1 | 1.55 | 42.6 | 24.3 | 11.9 |
| CSP3 | w/o-MR | 27.5 | - | 1.35 | - | 21.7 | - | 5.17 | - | 0.41 | - | 22.5 | - |
| | w/o-M | 56.3 | 104.7 | 1.80 | 33.2 | 21.8 | 0.37 | 17.1 | 231.0 | 0.73 | 78.3 | 25.3 | 12.7 |
| | EarnMore | 93.7 | 66.3 | 2.12 | 17.9 | 24.0 | 10.4 | 43.5 | 153.9 | 1.57 | 1151 | 21.18 | -16.23 |

**Effectiveness of Each Component (RQ1).** In Table 4, we study the impact of maskable stock representation, customizable stock pools, and re-weighting methods. Comparing EarnMore-w/o-M with EarnMore reveals a significant improvement due to the generalized pool-level maskable stock representation. The absence of this representation increases investment risk. Both GSP and CSPs benefit, with CSPs outperforming GSP, suggesting potential for higher returns with focused stock selection. Both GSP and CSPs benefit, with CSPs performing better, suggesting the potential for higher returns through a more focused stock selection.

Comparing the EarnMore-w/o-MR and EarnMore-w/o-M, we find that the re-weighting method can achieve significant improvements in profits by sparsifying portfolios, somehow in increasing

the MDD risk metric. Despite reducing the portfolio's diversity may decrease the chances of selecting stocks from various industries and potentially raise risks, especially in CSPs with limited industry variety, focusing on a select few industries can considerably improve returns and offset potential losses due to risk.

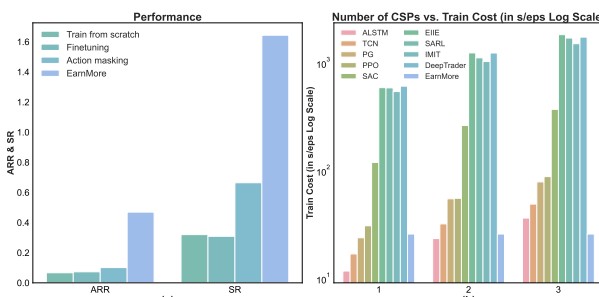

**Figure 5: (a) Comparing the performance of EarnMore with direct methods on DJ30. (b) Comparing the time costs of EarnMore with several other methods on DJ30.**

**Difficulties with Direct Methods (RQ2).** There are three simple approaches to transition from PM with GSP to PM with CSPs, which are *training-from-scratch*, *fine-tuning* and *action-masking*. The first thing to note that both the *training-from-scratch* and *fine-tuning* approaches lack the ability to make real-time adjustments to the stock pool during trading. Nevertheless, we make a comparison between these three methods and EarnMore in terms of DJ30 ARR and SR via SAC, with the metrics representing the average performance over three investment periods. As depicted in Figure 5(a), EarnMore significantly outperforms the other direct methods. It is worth noting that the challenges of *fine-tuning* and *training-from-scratch* may be closely related in the context of PM with CSPs, and that *action-masking* essentially relies on logits that do not accurately reflect the agent's actual decisions.

**Efficiency of EarnMore (RQ3).** Our framework is trained only once to meet the demand of various investors for customizable stock pools and individual decision making. As shown in Figure 5(b), we have selected several other methods to compare with EarnMore, and it can be demonstrated that as the number of CSPs increases, the efficiency of our framework shows up.

## 6 CONCLUSION AND FUTURE DIRECTION

This paper introduces a novel RL framework for portfolio management featuring adaptive investor preference and personal decision awareness for customizable stock pools. Maskable stock representation is enhanced by masking and reconstruction process, and a re-weighting method is introduced to improve sparsified portfolios. These improvements yield superior portfolio performance compared to the benchmark methods, as evidenced by various financial criteria. For future research directions, two key areas will be prioritized. Firstly, we acknowledge the need to enhance risk control, which will be addressed through the exploration of risk penalty optimization. Secondly, while our current approach utilizes a closed customizable stock pool, our aim is to develop a universal and open customizable stock pool that facilitates the easy addition or removal of stocks.

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

**Table 5: Technical indicators definitions and formulas.**

| Indicator | Calculation Formula |
|---|---|
| max_oc | $z_{max\_oc} = \max(p_t^o, p_t^c)$ |
| min_oc | $z_{min\_oc} = \min(p_t^o, p_t^c)$ |
| kmid2 | $z_{kmid2} = (p_t^c - p_t^o)/(p_t^h - p_t^l)$ |
| kup2 | $z_{kup2} = (p_t^h - z_{max\_oc})/(p_t^h - p_t^l)$ |
| klow | $z_{klow} = (z_{min\_oc} - p_t^l)/p_t^o$ |
| klow2 | $z_{klow} = (z_{min\_oc} - p_t^l)/(p_t^h - p_t^l)$ |
| ksft2 | $z_{sft2} = (2 \times p_t^c - p_t^h - p_t^l)/(p_t^h - p_t^l)$ |
| **Definitions** | $w \in W = [5, 10, 20, 30, 60]$ 
 $ret1 = (p_t^c - p_{t-1}^c)/p_{t-1}^c$ 
 $abs\_ret1 = Abs(ret1)$ 
 $pos\_ret1 = ret1 < 0?0 : ret1$ |
| roc | $z_{roc\_w} = Shift(p_t^c, w)/p_t^c$ |
| ma | $z_{ma\_w} = RollingMean(p_t^c, w)/p_t^c$ |
| std | $z_{std\_w} = RollingStd(p_t^c, w)/p_t^c$ |
| beta | $z_{roc\_w} = (Shift(p_t^c, w) - p_t^c)/(w \times p_t^c)$ |
| max | $z_{max\_w} = RollingMax(p_t^c, w)/p_t^c$ |
| min | $z_{min\_w} = RollingMin(p_t^c, w)/p_t^c$ |
| qtlu | $z_{qtlu\_w} = RollingQuantile[q = 0.8](p_t^c, w)/p_t^c$ |
| qtld | $z_{qtlu\_w} = RollingQuantile[q = 0.2](p_t^c, w)/p_t^c$ |
| rank | $z_{rank\_w} = RollingRank(p_t^c, w)/w$ |
| imax | $z_{imax\_w} = RollingArgmax(p_t^c, w)/w$ |
| imin | $z_{imin\_w} = RollingArgmin(p_t^c, w)/w$ |
| imxd | $z_{imxd\_w} = (RollingArgmax(p_t^h, w) - RollingArgmin(p_t^l, w))/w$ |
| cntp | $z_{cntp\_w} = RollingSum(ret1 > 0, w)/w$ |
| cntn | $z_{cntn\_w} = RollingSum(ret1 < 0, w)/w$ |
| cntd | $z_{cntd\_w} = z_{cntp\_w} - z_{cntn\_w}$ |
| sump | $z_{sump\_w} = RollingSum(pos\_ret1, w)/RollingSum(abs\_ret1, w)$ |
| sumn | $z_{sumn\_w} = 1 - z_{sump\_w}$ |
| sumd | $z_{sumd\_w} = 2 \times z_{sump\_w} - 1$ |

## A DETAILS OF DATASETS AND PROCESSING

In our experiments, we used stock data from Yahoo Finance, representing the U.S. stock markets. The evaluation included various stock indices and subsets, showcasing the robustness and real-world applicability of EarnMore. To demonstrate its versatility across different market scales, we intentionally selected two stock indices of contrasting sizes. This comprehensive evaluation validated the effectiveness and performance of EarnMore across diverse markets, making it suitable for various real-world scenarios.

The dataset consisted of historical daily k-line data for 10,273 U.S. stocks, spanning from the 1960s to 2022. The U.S. market was chosen for its global influence, long history, and abundance of stocks, commonly used in research studies. Our analysis focused on the period from September 2007 to June 2022, covering significant events like the subprime mortgage crisis, the COVID-19 pandemic, and geopolitical conflicts. To ensure data quality, we performed preprocessing to handle bad distributions, missing values, and extreme values, resulting in a dataset of 3,094 U.S. stocks with 95 technical indicators based on Qlib's Alpha158. For fair comparisons, we normalized the data using the mean and standard deviation of each stock in the training set, yielding a dataset of nearly 10 million data points. The definition and calculation process of technical indicators is shown in the Tab. 5. *Shift* and *Rolling* are both operations in pandas. $Shift(\cdot, w)$ is used to shift the index by a specified number of periods $w$. $Rolling[Agg](\cdot, w)$ is used to perform rolling calculations, which involves sliding a window $w$ over the data and applying a specific aggregation function $Agg$ to each window. For example, $Shift(p_t^c, w)$ actually is $p_{t-w}^c$ and $RollingMean(p_t^c, w) = \frac{1}{w}\sum_{i=0}^{w} p_{t-i}^c$.

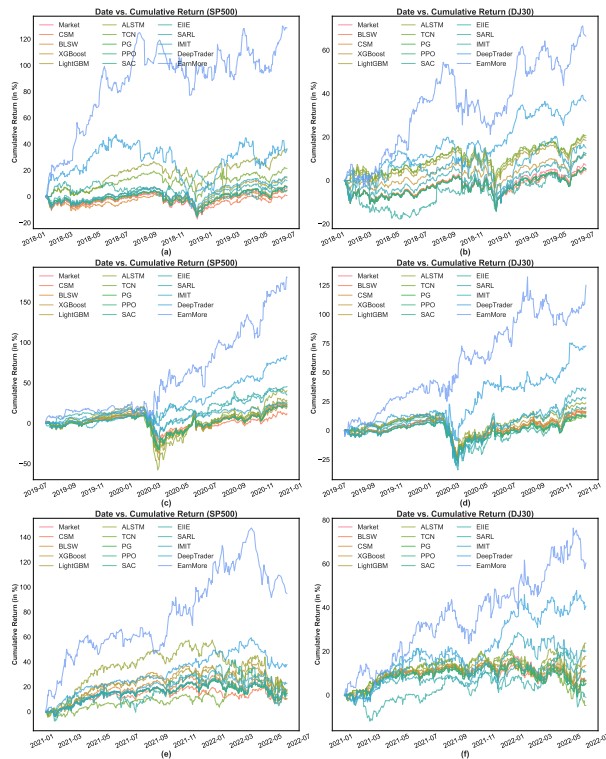

**Figure 6: Performance on GSP and CSPs for SP500 and DJ30**

## A.1 Evaluation Metrics

We compared EarnMore and baselines in terms of 6 financial metrics, including 1 profit criterion, 3 risk-adjusted profit criteria, and 2 risk criteria. Definitions and formulas are as follows:

- **Annual Rate of Return (ARR)** is the annualized average return rate, calculated as $ARR = \frac{V_T - V_0}{V_0} \times \frac{C}{T}$, where $T$ is the total number of trading days, and $C = 252$ is the number of trading days within a year. $V_T$ and $V_0$ represent the final and initial portfolio values.
- **Sharpe Ratio (SR)** measures risk-adjusted returns of portfolios. It is defined as $SR = \frac{\mathbb{E}[\mathbf{r}]}{\sigma[\mathbf{r}]}$, where $\mathbb{E}[\cdot]$ is the expectation, $\sigma[\cdot]$ is the standard deviation, $\mathbf{r} = [\frac{V_1 - V_0}{V_0}, \frac{V_2 - V_1}{V_1}, ..., \frac{V_T - V_{T-1}}{V_{T-1}}]^T$ denotes the historical sequence of the return rate.
- **Volatility (VOL)** is the variation in an investment's return over time, measured as the standard deviation $\sigma[\mathbf{r}]$.
- **Maximum Drawdown (MDD)** measures the largest loss from any peak to show the worst case. It is defined as: $MDD = \max_{i=0}^{T} \frac{P_i - R_i}{P_i}$, where $R_i = \prod_{i=1}^{T} \frac{V_i}{V_{i-1}}$ and $P_i = \max_{i=1}^{T} R_i$.
- **Calmar Ratio (CR)** compares average annualized return to maximum drawdown, assessing risk-adjusted performance. It is defined as $CR = \frac{\mathbb{E}[\mathbf{r}]}{MDD}$.
- **Sortino Ratio (SoR)** is a risk-adjusted measure that focuses on the downside risk of a portfolio. It is defined as $SoR = \frac{\mathbb{E}[\mathbf{r}]}{DD}$, where $DD$ is the standard deviation of negative return.

**Table 6: Performance comparison on SP500 and DJ30. Results in red, yellow and green show the best, second best and third best results on each dataset.**

| Stock Index | | SP500 | | | | | | DJ30 | | | | | |
|---|---|---|---|---|---|---|---|---|---|---|---|---|---|
| | | Profit | Risk-Adjusted Profit | | | Risk Metrics | | Profit | Risk-Adjusted Profit | | | Risk Metrics | |
| Pool | Strategies | ARR%↑ | SR↑ | CR↑ | SOR↑ | MDD%↓ | VOL↓ | ARR%↑ | SR↑ | CR↑ | SOR↑ | MDD%↓ | VOL↓ |
| GSP | Market | 9.320 | 0.556 | 0.702 | 17.120 | 26.160 | 0.014 | 6.710 | 0.458 | 0.776 | 15.560 | 22.200 | 0.013 |
| | BLSW | 11.630 | 0.696 | 0.894 | 21.450 | 24.560 | 0.013 | 7.610 | 0.512 | 0.857 | 16.930 | 21.540 | 0.012 |
| | CSM | 5.070 | 0.329 | 0.434 | 9.840 | 23.350 | 0.013 | 5.930 | 0.400 | 0.643 | 12.950 | 20.770 | 0.012 |
| | XGBoost | 10.690 | 0.377 | 0.473 | 13.650 | 19.300 | 0.016 | 10.260 | 0.343 | 0.599 | 10.420 | 14.760 | 0.013 |
| | LightGBM | 16.330 | 0.575 | 0.744 | 20.110 | 24.760 | 0.016 | 13.420 | 0.591 | 0.703 | 14.220 | 20.900 | 0.014 |
| | ALSTM | 43.50 | 1.157 | 1.367 | 22.501 | 35.820 | 0.026 | 15.030 | 1.186 | 0.590 | 14.890 | 28.070 | 0.013 |
| | TCN | 13.560 | 1.044 | 1.460 | 14.540 | 35.780 | 0.025 | 6.980 | 0.732 | 0.269 | 8.280 | 37.400 | 0.018 |
| | PG | 12.580 | 0.431 | 0.519 | 24.340 | 26.180 | 0.014 | 7.970 | 0.321 | 0.435 | 8.430 | 21.570 | 0.012 |
| | PPO | 15.130 | 0.537 | 0.742 | 14.770 | 24.100 | 0.013 | 9.240 | 0.385 | 0.512 | 10.140 | 20.810 | 0.012 |
| | SAC | 15.140 | 0.538 | 0.743 | 14.770 | 24.100 | 0.013 | 9.150 | 0.326 | 0.448 | 8.830 | 20.600 | 0.012 |
| | EIIE | 15.030 | 0.540 | 0.627 | 15.450 | 26.920 | 0.015 | 22.900 | 0.689 | 1.465 | 23.450 | 16.770 | 0.014 |
| | SARL | 21.240 | 0.756 | 0.970 | 21.230 | 24.000 | 0.013 | 21.920 | 0.786 | 1.109 | 23.020 | 20.400 | 0.012 |
| | IMIT | 50.300 | 1.162 | 1.949 | 35.050 | 25.420 | 0.018 | 27.640 | 0.909 | 1.593 | 27.380 | 20.050 | 0.014 |
| | DeepTrader | 60.290 | 1.980 | 2.195 | 34.260 | 28.580 | 0.013 | 32.230 | 1.335 | 1.440 | 27.110 | 21.190 | 0.013 |
| | **EarnMore** | 97.170 | 2.032 | 2.506 | 42.160 | 28.120 | 0.023 | 47.290 | 1.454 | 1.692 | 28.040 | 21.650 | 0.018 |
| | Improvement | 61.171% | 2.626% | 14.169% | 20.285% | - | - | 46.727% | 8.914% | 6.215% | 2.411% | - | - |
| CSP1 | Market | 16.581 | 0.722 | 1.242 | 26.728 | 28.703 | 0.017 | 10.784 | 0.599 | 1.203 | 23.793 | 21.869 | 0.013 |
| | BLSW | 16.446 | 0.714 | 1.230 | 26.328 | 28.655 | 0.017 | 11.078 | 0.607 | 1.217 | 24.030 | 21.919 | 0.013 |
| | CSM | 15.772 | 0.691 | 1.194 | 25.658 | 28.974 | 0.017 | 10.453 | 0.580 | 1.166 | 23.281 | 22.036 | 0.013 |
| | XGBoost | 17.265 | 0.371 | 0.582 | 15.720 | 22.988 | 0.017 | 10.293 | 0.339 | 0.605 | 6.953 | 14.390 | 0.013 |
| | LightGBM | 26.530 | 0.651 | 0.940 | 22.337 | 28.147 | 0.016 | 17.167 | 0.516 | 0.819 | 9.493 | 20.837 | 0.013 |
| | ALSTM | 18.710 | 0.796 | 0.817 | 18.002 | 36.762 | 0.025 | 22.786 | 1.212 | 0.765 | 19.502 | 23.875 | 0.013 |
| | TCN | 19.120 | 0.717 | 1.558 | 20.176 | 36.777 | 0.023 | 38.154 | 1.218 | 1.190 | 22.678 | 23.090 | 0.019 |
| | PG | 7.130 | 0.278 | 0.382 | 29.330 | 30.350 | 0.016 | 12.180 | 0.410 | 0.650 | 0.220 | 19.850 | 0.012 |
| | PPO | 22.200 | 0.557 | 0.817 | 15.150 | 28.330 | 0.016 | 12.070 | 0.404 | 0.634 | 10.810 | 20.060 | 0.012 |
| | SAC | 22.280 | 0.560 | 0.820 | 15.210 | 28.320 | 0.024 | 11.300 | 0.377 | 0.609 | 10.040 | 20.170 | 0.012 |
| | EIIE | 39.090 | 0.635 | 1.107 | 17.940 | 33.340 | 0.019 | 15.250 | 0.478 | 1.050 | 14.880 | 17.720 | 0.012 |
| | SARL | 34.330 | 0.820 | 1.159 | 22.330 | 27.860 | 0.016 | 24.140 | 0.638 | 1.047 | 17.450 | 22.470 | 0.014 |
| | IMIT | 34.030 | 0.793 | 1.241 | 22.620 | 27.950 | 0.018 | 27.740 | 0.757 | 1.251 | 21.490 | 21.350 | 0.014 |
| | DeepTrader | 20.973 | 0.860 | 1.516 | 31.222 | 28.389 | 0.017 | 20.071 | 0.920 | 1.781 | 33.901 | 22.203 | 0.015 |
| | **EarnMore** | 122.610 | 2.278 | 2.957 | 48.430 | 25.060 | 0.023 | 53.990 | 1.810 | 2.165 | 35.570 | 22.930 | 0.018 |
| | Improvement | 211.59% | 164.89% | 89.794% | 55.115% | - | - | 41.505 % | 48.604 % | 21.561 % | 4.923% | - | - |
| CSP2 | Market | 6.032 | 0.389 | 0.381 | 9.029 | 30.820 | 0.017 | 8.930 | 0.593 | 0.730 | 15.274 | 22.502 | 0.014 |
| | BLSW | 6.218 | 0.396 | 0.393 | 9.317 | 30.717 | 0.017 | 9.094 | 0.600 | 0.750 | 15.610 | 22.443 | 0.014 |
| | CSM | 5.924 | 0.379 | 0.367 | 8.660 | 30.584 | 0.017 | 8.388 | 0.572 | 0.705 | 14.701 | 22.329 | 0.014 |
| | XGBoost | 9.275 | 0.356 | 0.570 | 16.240 | 21.440 | 0.017 | 8.560 | 0.360 | 0.591 | 10.823 | 16.508 | 0.013 |
| | LightGBM | 10.687 | 0.383 | 0.506 | 18.060 | 30.120 | 0.016 | 13.610 | 0.561 | 0.831 | 16.717 | 20.553 | 0.012 |
| | ALSTM | 11.959 | 0.607 | 1.127 | 15.019 | 34.388 | 0.017 | 14.903 | 1.121 | 0.526 | 13.876 | 29.800 | 0.014 |
| | TCN | 12.192 | 0.456 | 0.408 | 10.729 | 36.163 | 0.024 | 31.249 | 1.186 | 1.079 | 19.243 | 27.634 | 0.019 |
| | PG | 7.030 | 0.275 | 0.378 | 29.290 | 30.370 | 0.016 | 8.970 | 0.366 | 0.546 | 10.510 | 20.700 | 0.012 |
| | PPO | 6.940 | 0.271 | 0.374 | 8.110 | 30.440 | 0.016 | 8.490 | 0.355 | 0.458 | 10.350 | 20.300 | 0.012 |
| | SAC | 6.930 | 0.272 | 0.374 | 8.130 | 30.450 | 0.016 | 8.750 | 0.366 | 0.566 | 10.720 | 20.180 | 0.012 |
| | EIIE | 22.260 | 0.848 | 1.491 | 26.630 | 25.170 | 0.016 | 14.180 | 0.595 | 1.062 | 18.800 | 25.890 | 0.017 |
| | SARL | 17.000 | 0.570 | 0.724 | 16.740 | 29.790 | 0.016 | 20.020 | 0.820 | 1.234 | 25.050 | 21.460 | 0.013 |
| | IMIT | 24.890 | 0.534 | 0.797 | 15.690 | 31.470 | 0.018 | 38.470 | 0.955 | 1.646 | 30.450 | 25.520 | 0.014 |
| | DeepTrader | 7.971 | 0.486 | 0.528 | 12.692 | 29.041 | 0.016 | 11.841 | 0.751 | 1.009 | 20.978 | 21.711 | 0.013 |
| | **EarnMore** | 110.110 | 2.279 | 3.116 | 46.990 | 25.770 | 0.024 | 43.400 | 1.549 | 1.744 | 28.010 | 24.330 | 0.019 |
| | Improvement | 342.39% | 168.75% | 108.79% | 60.430% | - | - | 46.727% | 12.815% | 30.607% | 5.953 % | - | - |
| CSP3 | Market | 7.048 | 0.497 | 0.578 | 14.183 | 24.235 | 0.013 | 1.068 | 0.115 | 0.329 | 6.730 | 25.023 | 0.013 |
| | BLSW | 7.493 | 0.535 | 0.628 | 15.348 | 23.723 | 0.012 | 1.292 | 0.133 | 0.357 | 7.298 | 24.748 | 0.013 |
| | CSM | 5.104 | 0.373 | 0.422 | 9.797 | 21.810 | 0.012 | 0.844 | 0.103 | 0.310 | 6.440 | 24.888 | 0.013 |
| | XGboost | 9.528 | 0.401 | 0.572 | 14.130 | 16.938 | 0.017 | -0.935 | 0.023 | 0.049 | 0.415 | 19.203 | 0.014 |
| | LightGBM | 12.863 | 0.512 | 0.710 | 19.083 | 22.863 | 0.012 | 3.907 | 0.185 | 0.288 | 5.080 | 23.440 | 0.013 |
| | ALSTM | 13.849 | 0.519 | 0.470 | 11.074 | 34.208 | 0.015 | 7.804 | 0.729 | 0.403 | 9.923 | 30.900 | 0.013 |
| | TCN | 9.733 | 0.854 | 1.184 | 16.498 | 34.958 | 0.023 | 7.228 | 0.570 | 0.401 | 10.098 | 36.633 | 0.018 |
| | PG | 8.970 | 0.370 | 0.456 | 22.570 | 24.230 | 0.013 | -0.550 | 0.005 | 0.038 | -0.300 | 23.520 | 0.012 |
| | PPO | 11.400 | 0.402 | 0.593 | 13.390 | 22.270 | 0.012 | -1.110 | -0.028 | 0.003 | -1.320 | 24.020 | 0.012 |
| | SAC | 11.420 | 0.487 | 0.732 | 13.420 | 22.270 | 0.012 | -0.750 | -0.005 | 0.020 | -0.610 | 23.740 | 0.012 |
| | EIIE | 17.590 | 0.742 | 1.068 | 20.390 | 21.260 | 0.012 | -4.050 | 0.022 | 0.057 | 0.650 | 29.930 | 0.016 |
| | SARL | 18.090 | 0.760 | 1.075 | 21.290 | 22.100 | 0.012 | 10.910 | 0.480 | 0.708 | 14.070 | 23.520 | 0.013 |
| | IMIT | 61.320 | 1.489 | 2.913 | 32.050 | 22.220 | 0.016 | 16.840 | 0.601 | 0.891 | 17.820 | 25.810 | 0.015 |
| | DeepTrader | 21.193 | 1.220 | 1.911 | 38.111 | 19.038 | 0.012 | 6.851 | 0.496 | 0.881 | 16.979 | 22.121 | 0.013 |
| | **EarnMore** | 93.670 | 2.120 | 2.720 | 42.810 | 24.030 | 0.022 | 43.460 | 1.572 | 1.969 | 29.670 | 21.180 | 0.018 |
| | Improvement | 52.756% | 42.377% | - | 12.330% | - | - | 158.08% | 115.64% | 89.675% | 66.498% | - | - |

## B  DETAILS OF IMPLEMENTATION

The dimensions of our state are $(B, N, D, F)$, where $B = 128$ represents the batch size, $N$ denotes the number of stocks. For DJ30, $N = 28$, and for SP500, $N = 420$. $D = 10$ represents the number of historical data days, and $F = 102$ represents the total number of features, including OHLC, technological indicators, and temporal information. It is worth mentioning that due to the large amplitude of Volume, which is not conducive to RL training, we have removed it from the features. For the *encoder*, *decoder*, *actor*, and *critic*, each of them consists of 2 layers of MLP with GELU activation function. The all components embedding dimension is 64.

All of our experiments were conducted on an Nvidia A6000 GPU, and we used grid search to determine the hyperparameters. The horizon length was chosen from the options $\{32, 64, 128, 256\}$, and we found that 128 yielded the best results. The batch size was set to 128, and the buffer size was $1e5$. The training process consisted of 2000 episodes using the AdamW optimizer. For both the MAEs component and the RL learning component of SAC, we utilized the Mean Squared Error (MSE) Loss function. During the grid search for the learning rates of the Actor, Critic, and MAEs component, we tested values within the range $\{1e-3, 1e-4, 1e-5, 1e-6, 1e-7\}$, and found that $1e-5$ yielded the best performance. The scheduler used was the multi-step learning rate scheduler with warm-up technique, starting with an initial learning rate of $1e-8$, which increased to $1e-5$ after 300 episodes, followed by subsequent multiplicative reductions by 0.1 at the 600th, 1000th, and 1400th episodes. The re-weighting parameters are given as $a = 0.6$, $b = 0.8$, $\mu = 0.7$, and $\sigma = 0.1$. The optimal temperature parameter $T$ is determined to be 0.1 from the set $\{10, 5, 1, 0.5, 0.1, 0.05, 0.01\}$. Default parameters were used for other baselines and all experiments ran with 3 seeds for computing average metrics.

## C  PSEUDOCODE FOR THE TRAINING AND INFERENCES PHASES OF EARNMORE

In this section, we present the detailed pseudocode for the training and testing phases of EarnMore. This includes the training phase with customizable stock pools simulated through masked token, as well as the inference phase involving an investor-customized target stock pool. Refer to Algorithm 1 and Algorithm 2.

## D  DETAILS OF COMPARISON WITH THE BASELINES

In this section, we conduct a comparative analysis of our method EarnMore in comparison to 14 benchmark models. The analysis is based on 6 key performance metrics applied to SP500 and DJ30 datasets. Specifically, these metrics include Average Rate of Return (ARR) as an indicator of portfolio performance, along with risk-adjusted measures like Sharpe Ratio (SR), Calmar Ratio (CR), and Sortino Ratio (SoR). Additionally, we consider risk-related metrics, Maximum Drawdown (MDD), and Volatility (VOL), to evaluate the risk implications of the strategies. The backtesting was conducted on 3 date splits, each with 3 seed values, and the reported metrics represent the averages derived from $3 \times 3$ experiments. For detailed information, please refer to Table ?? and Figure 6.

---

**Algorithm 1** Training of EarnMore

**Require:** Global Stock Pool $U$
**Ensure:** $\theta_1, \theta_2, \theta_e, \theta_c, \theta_{enc}, \theta_{dec}, \phi$  ▷ Parameters
   $s_t = [\mathbf{P}_t, \mathbf{Y}_t, \mathbf{D}_t]$  ▷ Initialize input data
   $\tilde{\theta}_1 = \theta_1, \tilde{\theta}_2 = \theta_2$  ▷ Initialize target network weights
   $\mathcal{D} = \Phi$  ▷ Initialize an empty replay buffer
   **for** each iteration and each environment step **do**
      $l_s = \psi_e(\mathbf{D_t}; \theta_e) + \psi_c(\mathbf{P}_t, \mathbf{Y}_t; \theta_c)$  ▷ Stock-level embedding
      $r = g(r; \mu, \theta, a, b)$  ▷ Sample a mask ratio
      $\tilde{l}_s = \eta_{mo}(l_s, r)$  ▷ Masking operation
      $l_p = \psi_{enc}(\tilde{l}_s; \theta_{enc})$  ▷ Pool-level embedding
      $\rho_t = \eta_{mf}(l_p, m)$  ▷ Maskable stock representation
      blue$a_t \sim Re(\pi_\phi(a_t|\rho_t), T)$  ▷ Sample action from the policy
      $s_{t+1} = p(s_{t+1}|s_t, a_t)$  ▷ Sample transition
      $\mathcal{D} \sim \mathcal{D} \cup \{(s_t, a_t, \rho_t, r(s_t, a_t), s_{t+1})\}$  ▷ Store transition
   **end for**
   **for** each gradient step **do**
      $\theta_i \leftarrow \theta_i - \lambda_Q \hat{\nabla}_{\theta_i} J_Q(\theta_i)$ for $i \in 1, 2$  ▷ Update Q network
      $\phi \leftarrow \phi - \lambda_\pi \hat{\nabla}_\phi J_\pi(\phi)$  ▷ Update policy network
      $\alpha \leftarrow \alpha - \lambda_\alpha \hat{\nabla}_\alpha J(\alpha)$  ▷ Adjust alpha
      $\tilde{\theta}_i \leftarrow \tau\theta_i + (1 - \tau)\tilde{\theta}_i$ for $i \in 1, 2$  ▷ Update target networks
      $\theta_e, \theta_c, \theta_{enc}, \theta_{dec} \leftarrow \lambda \hat{\nabla} J(\theta_e, \theta_c, \theta_{enc}, \theta_{dec})$
   **end for**
   **return** $\theta_1, \theta_2, \theta_e, \theta_c, \theta_{enc}, \theta_{dec}, \phi$  ▷ Optimized parameters

---

**Algorithm 2** Inference of EarnMore

**Require:** Global Stock Pool $U$, Customizable Stock Pools $CSPs = \{C_t|t = 1, 2, ..., T\}$  ▷ Input data
**Ensure:** $\{W_t|t = 1, 2, ..., T\}$  ▷ Portfolios of CSPs
   **for** each time step $t$ in $\{1, 2, ..., T\}$ **do**
      $s_t = [\mathbf{P}_t, \mathbf{Y}_t, \mathbf{D}_t]$  ▷ Initialize state
      $l_s = \psi_e(\mathbf{D_t}; \theta_e) + \psi_c(\mathbf{P}_t, \mathbf{Y}_t; \theta_c)$  ▷ Stock-level embedding
      Initialize mask index $M$ according to CSP $t$
      $\tilde{l}_s = \eta_{mo}(l_s, M)$  ▷ Masking operation
      $l_p = \psi_{enc}(\tilde{l}_s; \theta_{enc})$  ▷ Encode pool-level embedding
      $\rho_t = \eta_{mf}(l_p, m)$  ▷ Maskable stock representation
      $a_t = Re(\pi_\phi(a_t|\rho_t), T)$  ▷ Predict action from the policy
      $W_t \leftarrow a_t$
   **end for**

---

As shown in Table 6, our framework performance is evaluated in comparison to all other methods. EarnMore stands out by significantly improving its return potential across all datasets, while maintaining a minimal loss in risk control. In Figure 6, we have included comparative line diagrams of EarnMore and several other methods in terms of cumulative returns. It is shown that EarnMore demonstrates the best profit potential across all datasets.

As depicted in Figure 6(a) and 6(b), in October 2018, due to the impact of the U.S. monetary policy and economic confrontation, investor confidence in the stock markets is challenged, leading to a downward trend in the U.S. stock market during this period. EarnMore is impacted, resulting in a partial loss of returns, but it is anticipated to recover quickly, maintaining its overall superiority over other methods. The global COVID-19 pandemic reached its

highest point between February 14 and March 20, 2020, causing a significant economic downturn and raising serious concerns among investors. This resulted in a substantial decline in the stock market, with the SP500 and DJ30 indices dropping by 31.81% and 34.78%, respectively. As depicted in Figures 6(c) and 6(d), it's clear that EarnMore is much less affected in terms of returns compared to the other methods. Moreover, it continues to gain profits after the market starts to recover. Even during market downturns, EarnMore shows an ability to pinpoint stocks with the potential for higher

returns when the market bounces back. Starting in 2021, the U.S. stock market began its recovery, with the SP500 and DJ30 indices generally showing an upward trend. From Figures 6(e) and 6(f), it's evident that EarnMore, compared to other methods, is better at identifying stocks with upward momentum, maximizing returns. In March 2022, due to geopolitical conflicts, there was a brief downturn in the U.S. stock market. During this time, EarnMore is somewhat affected, showing noticeable changes in returns in the SP500, but still demonstrating a strong upward trend in the DJ30.

