# OpenReview forum: "Reinforcement Learning with Maskable Stock Representation for Portfolio Management in Customizable Stock Pools"
_ACM.org/TheWebConf/2024/Conference — TheWebConf24_

### Official Review · Reviewer_J4W2 · 2023-11-23

**Novelty:** 5
**Technical Quality:** 5

**Review:**

**Summary:**
The paper looks at the problem of Portfolio Management with customisable stock pools.
The paper proposes a reinforcement learning based method EarnMore which is trained once on the global stock pool and allows for CSPs.
The paper introduces learned masked token ([M]) which helps them treat the different stocks in the pool uniformly while identifying stocks that are less favorable.
The stocks are embedded to capture the different connections between stocks.

Existing works may need to be retrained from scratch for every combination of favored Stocks which can be time consuming.
Additionally, existing works loose the relationship between stocks if they were removed from GSP during the training process.
This work tackles these aforementioned challenges.

**Clarity:**
The paper is well written and is easy to read.

**Quality:**
The main technical contribution is to model CSPs into the RL framework by adding learned masked token and reweighting the allocation based on CSP preference.
The paper performs extensive experiments with 14 baselines.

**Significance:**
The paper introduces the notion of CSPs to PM.
The problem tackled is incremental but important in its own right.

**Pros:**

- The experiments are extensive with many baselines.

- The key contribution has been to envision the pipeline tailored for this work. Some of the individual parts of the pipeline were modelled based on existing works.

- The paper is well written and is easy to read.

**Cons:**

- The experiments consider numerous baselines and two different stock exchange data. It would be great to have one more dataset from a different stock exchange.

- While experiment section is detailed and extensive, adding a few scenarios could strengthen the paper (see detailed comments below).

**Questions:**

1. While the experiments considered many baselines and competitors, it would be great to add one more dataset. Adding a dataset like NASDAQ 100[23], HSI[30], CSI100[30] would strengthen the paper.

2. In "Performance on Customizable Stock Pools" of 5.5 Page 7: Some scenarios of changes of CSP have been explored. While these paint a picture of what can potentially happen when a trader decides to mark a stock as unfavorable or viceversa, it would be good to instead model human behavior of changes in CSP and then provide statistics of on changes in worth of portfolio.

3. Additionally, plots 5a and 5b show the performance of EarnMore with competitors with different CSPs. It would also be great to comapre EarnMore with competitors when CSP has changed, i.e. retrain the competitors with new set of stocks in portfolio and compare them with EarnMore. Unlike the competitors, EarnMore can still deduct certain information of relation between stocks even when a stock is not in CSP. Hence, it would be interesting to see how it fares.

4. Minor:
  - Colors of Plot 5.a., 5 b are very similar and not distingushable. It would be great to use contrasting colors.
  - Page 1 : "Therefore, We" - > "Therefore, we"
  - Appendix D last line - hanging reference to Table ??

**Reviewer Confidence:**

3: The reviewer is confident but not certain that the evaluation is correct

**Scope:**

3: The work is somewhat relevant to the Web and to the track, and is of narrow interest to a sub-community

---

### Official Review · Reviewer_wXZ7 · 2023-11-25

**Novelty:** 4
**Technical Quality:** 5

**Review:**

The paper proposes to use reinforcement learning with learnable and maskable stock representation to perform portfolio management on customized stock pools (instead of the traditional global pool, which can be computationally expensive). The approach enables the unified representation of stock pools without the need to learn a model for each combination. Specifically, a masking and reconstruction process is used to learn the customized pool-level embedding. A reweighing mechanism is then used to rescale the distribution of portfolios to concentrate on favorable stocks and neglect stocks outside the target pool. Experiments are conducted on real stock market data, SP500 and DJ30, and EarnMore performs better than rule-based, ML-based, and DL-based baselines.

Comments:
Overall, the paper solves a practical problem in portfolio management and the method could potentially applied to other settings that involve combinatorial action spaces.

One of my questions/concerns is on how well the method can scale and generalize when the general pool's dimension is very large (e.g., SP500) while the number of stocks in the customized pool is very small (e.g., three stocks)? It would be helpful for the readers and practitioners if some advice can be provided on when to train a one-time general model and when to train a smaller customized model.

Minor:
line 201: "self-self-supervision"

**Questions:**

See above.

**Reviewer Confidence:**

3: The reviewer is confident but not certain that the evaluation is correct

**Scope:**

3: The work is somewhat relevant to the Web and to the track, and is of narrow interest to a sub-community

---

### Official Review · Reviewer_G2Ue · 2023-11-30

**Novelty:** 4
**Technical Quality:** 5

**Review:**

## Paper Summary
The paper proposes a new way for variable and Customizable Stock Pool (CSP) in the portfolio management when the Global Stock Pool (GSP) is set. It uses masked Auto-encoders to compress stock price and feature into masked stock representation. This operation strengthens the connections between stocks in the GSP through the masking and reconstruction. A soft Actor Critic is then employed on the masked stock representation to maximize total portfolio value. A re-weighting scheme then zero out stocks with small weights. Extensive experiments on S&P500 and DowJones30 are performed to demonstrate the superior performance of the proposed method.

## Strength
Masked auto encoder is not new, but applying it to portfolio management helps extract correlation across different stocks. The proposed method enables one-time training and cater for different customizable stock pool selected by different investors. The experiments are strong and extensive with Ablation study, demonstrating much better performance on the Global Stock Pool (GSP). Superior performance is also demonstrated on the three CSP.

## Weakness
Weaknesses can be divided into two categories
1. What to do in test time is not clearly explained. Is any masking performed on test time? If so, why should we forsake useful features?
2. The provided methodology of Masked auto encoder can only **represent** unfavorable stocks outside of customizable stock pool, but cannot enforce the Actor Critic to put low weight on it. So portfolio manager can still end up with stocks masked out, even with re-weighting.

For details see **bolded** questions in the next section.

## Summary of Recommendation
Though the paper provides state of the art performances and strong experiments results, the two weaknesses are too important to be left unaddressed. The review is subject to change when successfully addressing these questions.

## After responses to the review
Happy with the two main points above. Scores for technical quality updated accordingly.

**Questions:**

## Question for Section 3.2
2. For RL rewards of each time step, what is the advantage of using portfolio value compared to changes in portfolio value? An RL that learns to only hold cash can still have constant reward each time step.

## Question for Section 4
1. **The first main contribution claimed by the paper is "a learnable masked token". Currently, it is not fully discussed. Is it the same with that in [1] and [2] where, "mask token is a shared, learned vector that indicates the presence of a missing part to be predicted"?**
2. **The Actor Critic uses the maskable stock representation as input. However, nothing is penalizing the actor-critic from putting weights on the masked stocks. Re-weighting only zeros out small weights but does not prevent policy network from putting weights on masked stocks in the first place. And there is currently no projection steps**.
3. **Should re-weighting be included during RL optimization? It is clearly differentiable and affects the rewards achieved.**
4. **The paper does not describe how to run the masking and actor critic during the test time. Figure 2) module a) shows some stocks can be selected by the investors and direct masking operations are performed. Is this performed on test time? If so, features of stocks outside of CSP won't go into the Actor Critic. Why should we forsake the stock features outside the CSP? Extra information should always help.  If not how to represent CSP in test time?**
5. Randomized masking parameter r is sampled from a truncated normal distribution. In practice, investors typically looks at stocks from a same industry collectively. Maybe correlated masking can further improve the performances?
6. RL Optimization order: Q-value, alpha, strategy network and finally maskable stock representation. One would intuitively think the maskable stock representation should be trained first, and then followed by Actor Critic, as the latent space vectors that can fully reconstruct the original space be important features for the Actor Critic to act upon.

[1] He, Kaiming, et al. "Masked autoencoders are scalable vision learners." Proceedings of the IEEE/CVF conference on computer vision and pattern recognition. 2022.
[2] Devlin, Jacob, et al. "Bert: Pre-training of deep bidirectional transformers for language understanding." arXiv preprint arXiv:1810.04805 (2018).

**Reviewer Confidence:**

3: The reviewer is confident but not certain that the evaluation is correct

**Scope:**

3: The work is somewhat relevant to the Web and to the track, and is of narrow interest to a sub-community

---

### Official Review · Reviewer_GMFm · 2023-12-01

**Novelty:** 4
**Technical Quality:** 5

**Review:**

The paper studies portfolio management(PM) using reinforcement learning, where the stock pools are customizable (CSP).  A naive approach requires retraining RL agents even with a tiny change in the stock pool, which leads to high computational costs.  They propose a rEinforcement leARNing framework with Maskable stOck REpresentation to handle PM with CSPs through one-shot training in a global stock pool (GSP).  Specifically, they use masking to remove the stocks outside the target pool, and learn stock representations through a self-supervised masking and reconstruction process.  Finally, they use reweighting to focus the portfolio on favorable stocks and neglect the stocks outside the target pool.  They did several experiments and showed their method (EarnMore) significantly outperforms 14 state-of-the-art baselines in terms of six financial metrics with over 40% improvement in profit.

Opinion:
I am only knowledgeable in this area, and will delegate technical comments to other reviewers.  The paper is well written, and the simulation result has impressive improvement over previous methods, e.g. achieving the highest Annual Rate of Return of 97%.  However, the paper only considers a small number of customizable stock pools (three?).  They may test their method on more choices of CSPs (e.g., collections of random k stocks).  In particular, I wonder if the intuition exploiting the relationship between stocks is still feasible for more adversarial choices of CSPs.

**Questions:**

See above.

**Reviewer Confidence:**

1: The reviewer's evaluation is an educated guess

**Scope:**

3: The work is somewhat relevant to the Web and to the track, and is of narrow interest to a sub-community

---

### Decision · Program_Chairs · 2024-01-22

**Decision:**

Accept

**Comment:**

Most of the reviews are positive about this paper, and the author carefully addressed all reviewers' questions. I will recommend (weak) acceptance.